# Volumetric imaging of human mesenchymal stem cells (hMSCs) for non-destructive quantification of 3D cell culture growth

Oscar R. Benavides[1]*, Holly C. Gibbs[1,2], Berkley P. White[1], Roland Kaunas[1], Carl A. Gregory[3], Alex J. Walsh[1], Kristen C. Maitland[1,2¤]

**1** Department of Biomedical Engineering, Texas A&M University, College Station, Texas, United States of America, **2** Microscopy and Imaging Center, Texas A&M University, College Station, Texas, United States of America, **3** School of Medicine, Texas A&M Health Science Center, Bryan, Texas, United States of America

¤ Current address: Imaging Program, Chan Zuckerberg Initiative, Redwood City, California, United States of America

* obenavides@tamu.edu

**Data Availability Statement:** The original data in the paper is available from the Cell Image Library repository (direct link: http://cellimagelibrary.org/

## Abstract

The adoption of cell-based therapies into the clinic will require tremendous large-scale expansion to satisfy future demand, and bioreactor-microcarrier cultures are best suited to meet this challenge. The use of spherical microcarriers, however, precludes in-process visualization and monitoring of cell number, morphology, and culture health. The development of novel expansion methods also motivates the advancement of analytical methods used to characterize these microcarrier cultures. A robust optical imaging and image-analysis assay to non-destructively quantify cell number and cell volume was developed. This method preserves 3D cell morphology and does not require membrane lysing, cellular detachment, or exogenous labeling. Complex cellular networks formed in microcarrier aggregates were imaged and analyzed *in toto*. Direct cell enumeration of large aggregates was performed *in toto* for the first time. This assay was successfully applied to monitor cellular growth of mesenchymal stem cells attached to spherical hydrogel microcarriers over time. Elastic scattering and fluorescence lightsheet microscopy were used to quantify cell volume and cell number at varying spatial scales. The presented study motivates the development of on-line optical imaging and image analysis systems for robust, automated, and non-destructive monitoring of bioreactor-microcarrier cell cultures.

## Introduction

In 2002, the Food and Drug Administration (FDA) announced a new science-based initiative to modernize quality management of pharmaceutical manufacturing and product quality by developing and implementing technologies that measure, control, and/or predict quality and performance of a process or product [1]. Utilizing Quality by Design (QbD) principles, traditional pharmaceutical and cell-based therapy manufacturers have recently been encouraged to develop and utilize Process Analytical Technologies (PATs) that perform real time or near real

groups/54864) with DOI (doi:10.7295/W9CIL54864).

**Funding:** This work was supported by the Texas A&M President's Excellence Fund (CG, RK, KM; Award #: 76; https://pef.tamu.edu/xgrants/index.html) and the Silicon Valley Community Foundation through the Chan Zuckerberg Initiative CZI Imaging Scientist Program (HG; Award #: 2019-198168; https://chanzuckerberg.com/). The funders had no role in study design, data collection and analysis, decision to publish, or preparation of the manuscript.

**Competing interests:** The authors have declared that no competing interests exist.

time monitoring of key variables throughout the manufacturing process to ensure quality of the final pharmaceutical product [2, 3]. In cell-based therapy manufacturing, PATs are designed to increase understanding of the cell culture process and facilitate the monitoring and control of critical process parameters (CPPs) that directly influence the quality and safety of the final cell product [4]. Ideal PATs operate in on- or in-line configurations, permitting automated and (near-) real time non-destructive measurements and analysis. The identification of CPPs for stem cell cultures and the development and deployment of PATs will improve the ability to study novel cell lines and expansion methods and enhance the capability to monitor, control, and ultimately predict the final product quality [2, 5].

Cell-based therapies, or cytotherapies, have the potential to address an unmet need for therapies that cure or treat chronic diseases such as cancer, osteoporosis, diabetes, and stroke [6–9]. Entry into the clinic will require billions of cells per indication per year, and one critical challenge in upstream cytotherapy manufacturing is the efficient large-scale expansion of stem cells to maximize yield while maintaining safety and therapeutic efficacy [10]. Even with the latest generation of multi-stacked cell factories, two-dimensional (2D) monolayer cultures have limited surface area for expansion, are labor- and reagent-intensive, and require serial passaging, which renders them sub-optimal for large-scale cellular expansion [11]. Three-dimensional (3D) cell cultures better mimic the *in vivo* stem cell niche than standard monolayer cultures while exploiting the 3rd spatial dimension for cellular expansion [12–14]. Bioreactor-microcarrier suspension cultures are the most promising 3D cell culture method as bioreactors can be scaled up to volumes of 1,000 liters, and microspheres greatly increase the available surface area to volume ratio, reduce labor and reagent use, and can be functionalized for specific needs [10], [15–18]. Furthermore, several groups have shown that 3D microcarrier human mesenchymal stromal cell (hMSC) cultures can provide a greater cell yield than 2D monolayer cultures without compromising viability, identity, or differentiation potential [19–23]. Recently, our group used spherical biodegradable gelatin methacryloyl (gelMA) microcarriers and bioreactor suspension cultures to demonstrate scalable expansion, rapid harvest, and non-destructive 3D *in toto* sub-micron visualization of induced pluripotent stem cell-derived hMSCs (ih-MSCs) via reflectance confocal microscopy (RCM) without the need for detachment from the microcarrier surface [24].

In regenerative medicine and biopharmaceutical manufacturing involving expansion and harvest of living cells, cell number is the most fundamental cell culture process parameter that requires quantification. Cell enumeration is needed to evaluate viability and proliferation, and in functional assays where activity is normalized to cell number such as engraftment [25, 26]. While there is no single established cell enumeration method for microcarrier cultures, essentially all off-line methods are destructive as they require either detachment of cells from the microcarrier surface and/or membrane lysing and exogenous labeling [27]. Two of the most common off-line cell enumeration and viability assays, trypan blue dye exclusion and live/dead fluorescence using Calcein AM and propidium iodide (PI), both of which are based on membrane integrity, are destructive to the samples and remove potentially valuable morphological and spatial distribution information [28–34]. Additionally, cell number can be measured on-line or in-line via a number of optical techniques, such as optical density measurements, *in situ* microscopy, micro-flow imaging, imaging and flow cytometry, and IR and fluorescence spectroscopy, and non-optical methods based on dielectric spectroscopy, acoustic measurements, or the chemical analysis of off-gas, media, protein, or DNA content in a sample [35–53].

Non-visualization cell enumeration methods are incapable of providing insight on cell morphology or spatial distribution, which in traditional monolayer cultures are readily monitored using in-process brightfield or phase-contrast microscopy; these CPPs are informative

and potentially predictive features of cellular fate, proliferation, and functional potential in monolayer cultures [4, 5, 30–34, 54, 55]. The evaluation of microcarrier surface confluency and spatial distribution of cells can provide insight into the culture microenvironment and better enable automated, objective real-time release of intermediate upstream cell cultures once a certain confluency threshold is reached [30, 56]. An ideal biomass monitoring PAT for bioreactor-microcarrier anchorage-dependent cell culture performs measurements on-line or in-line and *in toto*, leaving cells attached to microcarriers and cell-microcarrier aggregates undisturbed so as to preserve cell morphology and 3D spatial distribution information.

Several imaging, microscopy, and visualization methods for cell enumeration in microcarrier-based cultures have been investigated, but an industry standard has yet to be determined. Automated image analysis could be incorporated into image- or visualization-based assays for more rapid and robust quantification [57–59]. Trypan blue dye exclusion and live/dead fluorescence labeling both require exogenous contrast agents, so they cannot be incorporated into on- or in-line assays. Off-line fluorescence-based direct cell enumeration assays are, however, used to correlate experimental on-line cell enumeration or biomass sensors [35, 60]. These assays have been based on total fluorescence intensity as opposed to 3D spatial volume which considers 3D cell morphology. Volumetric fluorescence microscopy can be used to characterize cell density, distribution, and morphology, but requires destructive exogenous fluorescent markers [61, 62]. We previously demonstrated RCM, based on back-scattered elastic photons, could be employed to achieve label-free, sub-micron *in toto* visualization of hMSCs attached to spherical microcarriers [24]. This optical method allows for cell enumeration, but raster scanning a 3D volume of ~$150^3$ $\mu m^3$ is too slow for on-, in-, or even off-line measurements. The rapid and photo-efficient light sheet microscopy (LSM) technique presents a more viable method for non-destructive monitoring of microcarrier cultures [63, 64]. Fortunately, elastic scattering light sheet microscopy (esLSM), also known as light sheet tomography (LST), can be used for *in toto* imaging of hMSCs attached to microcarriers and quantification of cell number while preserving cell morphology. Contrast is generated from elastically scattered photons as opposed to more traditional light sheet fluorescence microscopy (LSFM) that utilizes fluorescence for imaging [65–67].

Here, we report a proof-of-concept study on volumetric optical imaging and semi-automated image analysis for off-line fluorescence and on-line elastic scattering quantification of cell number and volume of hMSCs cultured on spherical hydrogel microcarriers *in toto*, without the need for cellular detachment. The off-line fluorescence assay utilizes LSFM and Cell-Tracker Green cytoplasmic and DRAQ-5 nuclear labeling for cell volume quantitation and direct cell enumeration of single microcarriers and large aggregates. This fluorescence assay is the first imaging-based assay to use volumetric data to more accurately characterize the 3D microcarrier cell culture and the first to directly enumerate cells within large aggregates *in toto*. The on-line assay utilizes esLSM and image analysis (ELIAS) to quantify cell volume of single microcarriers and aggregates non-destructively. Cell number from the fluorescence assay was correlated to cell volume from the ELIAS assay. The ELIAS assay has the capability to be adopted as an on-line PAT for robust non-destructive monitoring of bioreactor-microcarrier cell culture growth which would improve process and quality control for cytotherapy manufacturing.

## Material and methods

### Induced pluripotent stem cell-derived hMSC (ih-MSC) culture

Passage 4 ih-MSCs were first expanded in low-density monolayer cell culture in complete culture medium (CCM) (α-Minimum Essential Medium, 10% fetal bovine serum, 2 mM L-

glutamine, 100 U/mL penicillin, and 100 μg/mL streptomycin) to obtain the required cell numbers. The ih-MSCs [68] were cultured in a rotating wall vessel (RWV) bioreactor (RCCS-8DQ bioreactor (Synthecon, Houston, TX) fitted with 10 mL high aspect ratio vessels [69] on custom-fabricated 120 ± 6.2 μm diameter gelMA microcarriers [24]. For this purpose, approximately 110,000 gelMA microcarriers with a combined growth area of 50 cm$^2$ and 5x10$^4$ cells (1000 cells/cm$^2$) were incubated in 10 mL of CCM in the RWV bioreactor at 24 revolutions per minute. Half of the media was replaced with fresh CCM every 2 days. Specimens were recovered and fixed at passages 4 and 7 on days 3 and 7, for a total of four samples.

## Sample preparation

At day 3 and day 7 of RWV bioreactor culture, CCM was removed and microcarrier-expanded cells were suspended in 1 mM concentration of CellTracker Green (CTG) for 45 minutes. The CTG target is distributed uniformly in the cell cytoplasm, and was used here to visualize 3D cell morphology and quantify cell volume. Cells were fixed with 4% paraformaldehyde (PFA) and stored in phosphate buffered saline (PBS) at a concentration of 3 mg particles/mL PBS for long-term storage. Fixed microcarrier-cell samples were incubated with a 5 μM DRAQ-5 DNA and 6.5 μM DiI plasma membrane staining buffer at 37º C for 30 minutes with agitation, then rinsed with PBS. The far-red fluorescent DRAQ-5 stain was used for cell nuclei visualization and direct cell enumeration. The orange fluorescent DiI label was used to illustrate a simpler staining method for visualization of the plasma membrane only.

The microcarriers were embedded in 1% agarose in a custom-designed and 3D-printed sample chamber (S1 Fig) [70]. A 300 μL aliquot was loaded into each sample chamber at a concentration of 6 mg particles/mL agarose. The chamber enables dual-sided lightsheet illumination, trans-illumination for widefield imaging, and >180˚ sample rotation for multi-view acquisition and optimized sample positioning.

## Off-line fluorescence-based cell enumeration and cell volume quantification

The Zeiss Z1 Lightsheet microscope, with a 20X 1.0 NA (water) detection objective lens and 10X 0.2 NA illumination objective lenses, was used for *in toto* imaging of fixed ih-MSCs attached to spherical microcarriers. The 488 nm (power 5%) and 638 nm (power 9%) lasers were used to excite the CTG and DRAQ-5 fluorophores, respectively. The voxel size was 0.2 x 0.2 x 0.45 μm$^3$ to satisfy Nyquist sampling requirements. The emission filters used were 505–545 nm and 660+ nm for CTG and DRAQ-5, respectively. Dual objective illumination with pivot scanning and online max fusion was used to improve illumination of microcarrier aggregates and reduce acquisition time. The camera integration time was set to 20 ms per frame, or 50 FPS, and the illumination power was adjusted to use the full dynamic range of the detector.

Imaris image analysis software was used to view and analyze the 3D volumes. For direct cell enumeration, the 'Spot' function was used on the DRAQ-5 volumes with an object size filter of 10 μm for automatic detection of cell nuclei, followed by a manual high-pass intensity threshold to further segment out cell debris and ultimately enumerate only cell nuclei (S2 Fig). For quantification of cell volume, the 'Surface' function was used on the CTG volumes with a manual high-pass intensity threshold and size filter to remove cell debris from quantification (S3 Fig). No preprocessing of the data was required for cell segmentation as the gelMA microcarriers produce little background signal [24]. For enumerating microcarriers, the 'Surface' function was used on the CTG volume with a low-pass intensity-based threshold to create a solid object. Then, the 'Spot' function with a 90 μm size filter was used to automatically enumerate individual spherical microcarriers.

### On-line elastic scattering-based cell volume quantification

The agarose-embedded and mounted microcarrier-cell samples were used to evaluate the feasibility of the ELIAS method for label-free, non-destructive, *in toto* imaging and characterization of 3D microcarrier cell culture growth.

For elastic scattering imaging on the Z1 Lightsheet microscope, the 638 nm laser (power 0.1%) was used to illuminate the sample. The laser blocking filter and emission filters were removed from the optical path. The camera acquisition time was minimized to 10 ms per frame, and the laser power was adjusted until there were no saturated pixels when viewing a cell. Dual-sided illumination, pivot scanning, and online max fusion were turned on. The Imaris 'Surface' function was used to segment the cells and microcarriers from each other and the agarose. For cell volume quantification, a high-pass intensity threshold and a high-pass size filter were used (S4 Fig). For microcarrier enumeration, a low-pass intensity threshold and 90 μm size filter were used (S5 Fig).

## Results

### 3D visualization via lightsheet microscopy

To demonstrate the ability to use volumetric fluorescence and label-free elastic scattering microscopy for direct and non-destructive quantitative monitoring of cell culture growth, ih-MSCs attached to gelMA microcarriers were imaged at four timepoints using lightsheet microscopy (Fig 1). The gelMA microcarriers, which have a refractive index (n) of 1.35 [24], permit 3D visualization of the entire microcarrier surface and core, enabling direct cell enumeration *in toto* via DRAQ-5-labeled nuclei and quantification of cell volume using LSFM (S1 and S2 Videos) and esLSM (S3 and S4 Videos) while preserving the integrity of the cell morphology.

The CTG volumes reveal the individual cell morphologies, and there appeared to be an increase in total cell volume from day 3 to day 7 in both passages (Fig 1a–1d). The label-free elastic scattering shows similar cell morphology and cell growth over time as the fluorescence data (Fig 1e–lh). This suggests both contrast methods can be used to view cells and quantify cell volume. The DRAQ-5-labeled nuclei data show an increase in cell density from day 3 to day 7 within each passage (Fig 1i–1l). A maximum of 19 cells per microcarrier was observed at Passage 4 Day 7 using the DRAQ-5 data (Fig 1j). The DRAQ-5 volumes were used to monitor the distribution of cells/microcarrier on single microcarriers over time (Fig 1m). Aggregates were excluded from the histograms to avoid weighting against single microcarriers. Similarly, non-populated microcarriers were not studied. A zero-truncated Poisson distribution was fit to the histograms to account for excluding empty microcarriers from acquisition and analysis [71]. These data suggest a decrease in cells/microcarrier or cell density at passage 7 compared to passage 4 overall. The elastic scattering signal seems to originate from the cytoplasm, and the nucleus tends to appear as a cavity that exhibits little to no elastic scattering signal (Fig 1h). This cytoplasm-dominant elastic scattering phenomena has been previously reported [72], and here we similarly show that nuclear-bound fluorescent markers and elastic scattering microscopy provide complimentary information on different cell regions.

### Optical sectioning and hydrogel microcarriers

In LSM, a stack of thin (~2 μm) planes is sequentially illuminated within the microcarrier sample, allowing more precise localization of interesting higher-resolution biological phenomena, such as a cell infiltrating the center of a gelMA microcarrier (Fig 2). The infiltration is clearly discernible in a 3D rendering of the dataset using both fluorescence and elastic scattering

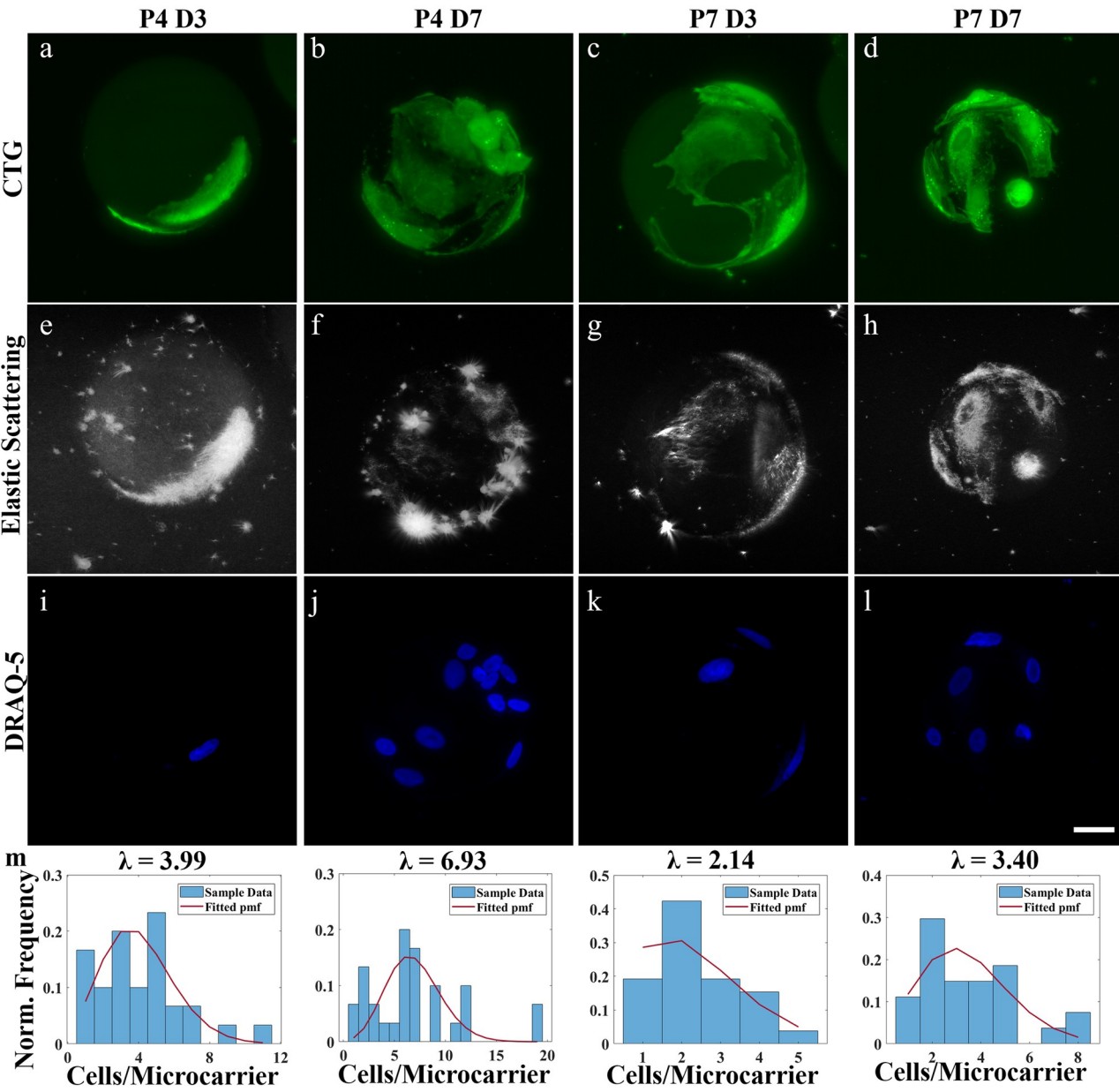

**Fig 1. LSM and hydrogel microcarriers enable direct cell enumeration (DRAQ-5) and volume (CTG) quantification *in toto*.** Representative 2D LSM max. intensity projections of ih-MSCs attached to gelMA microcarriers at passage 4 day 3 (P4 D3), passage 4 day 7 (P4 D7), passage 7 day 3 (P7 D3), and passage 7 day 7 (P7 D7) using **a-d)** CTG fluorescence and **e-h)** elastic scattering contrast. **i-l)** DRAQ-5 labeled nuclei were used to estimate the **m)** normalized frequency of counted ih-MSCs attached to single gelMA microcarriers at P4 D3 (n = 30), P4 D7 (n = 20), P7 D3 (n = 23), and P7 D7 (n = 26). A zero-truncated Poisson distribution is fit over the sampled data (red line). Scale bar = 25 μm.

contrast (S5 and S6 Videos). In the 2D max. intensity projection of the microcarrier 3D volume from the LSM, it is not possible to discern the cell process burrowing into the core of the microcarrier (Fig 2a). This appears to be a large, binucleated cell wrapping around ~1/3 of the microcarrier (Fig 2b). A max. intensity projection of the middle 1/3 volume of the microcarrier allows for visualization of the cell process extending into the core of the microcarrier (Fig 2c). There is, interestingly, a clear delineation of the nuclear envelope and microcarrier infiltration outlined by DiI staining (Fig 2d) [73]. Assuredly, the elastic scattering mode is also able to

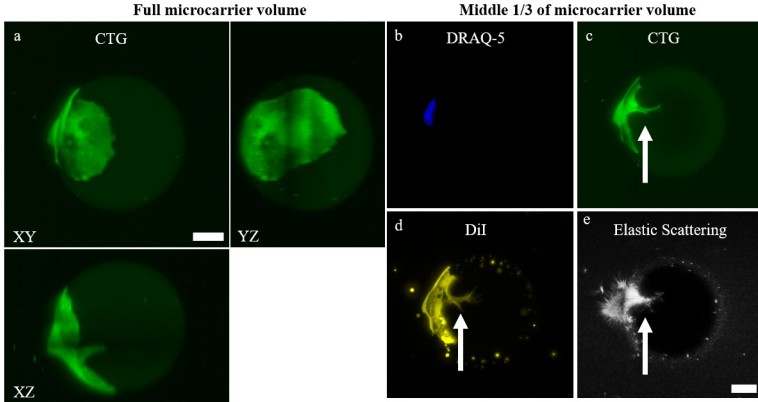

**Fig 2. The optical sectioning capabilities of LSM permits visualization into the hydrogel microcarrier. a)** 2D max. intensity and orthogonal projection of P7 D3 microcarrier with a single CTG-expressing cell. 2D max. intensity projections using the middle 1/3 of the volume to view the interior of the microcarrier using **b)** DRAQ-5, **c)** CTG, **d)** DiI plasma membrane stain, and **e)** elastic scattering contrast. Owing to the optical sectioning capabilities of LSM and refractive index matching of the microcarriers, biological features such as cell infiltration into the microcarrier can be visualized (white arrow). Scale bar = 25 μm.

visualize this cell infiltration, illustrating that complex cell-microcarrier interactions and cell morphologies can be visualized and characterized with esLSM (Fig 2e).

## *In toto* imaging of large aggregates

Owing to the superior optical properties of the hydrogel microcarrier and optical sectioning ability of LSM, large cell-microcarrier aggregates that form later in culture with increased cell growth can reach volumes $> 4$ mm$^3$ and still be imaged *in toto*. This permits semi-automatic cell enumeration, microcarrier enumeration, and cell volume quantification of large microcarrier aggregates using both off-line fluorescence and the ELIAS methods (Fig 3). The CTG data shows a complex network of cellular connections throughout the aggregate (Fig 3a). The DiI plasma membrane stain, which does not require a live incubation period for conversion into a fluorescent marker, similarly reveals a large cellular network (Fig 3b). There were 5,673 individual cell nuclei enumerated using the DRAQ-5 data (Fig 3c). Using the elastic scattering data, which provides slight contrast for the gelMA material, 1,754 microcarriers were detected, for an average of 3.32 cells per microcarrier (Fig 3d). The elastic scattering modality also reveals the cells throughout the entire microcarrier aggregate (Fig 3d). Small scatterers in the agarose and cell debris on the microcarrier surfaces can be segmented out with intensity- and size-based filters as cells scatter at higher intensity values and are larger than the debris (Fig 3e). The higher-resolution, merged projection of the DRAQ-5 and CTG data illustrates the density of cells within aggregates (Fig 3f). The elastic scattering and DRAQ-5 zoomed-in merged projection reveals similar cell morphologies as the CTG data even for this large aggregate (Fig 3g). A 1 mm sweep in depth through the aggregate further exemplifies that elastic scattering can visualize both microcarriers and the complex network of cellular connections that create an aggregate in culture (S7 Video).

## Cell enumeration and cell volume quantification

An off-line fluorescence-based assay for direct cell enumeration and cell volume quantification of cell expansion on microcarriers was developed. This method is based on the volume of cellular fluorescence as opposed to total fluorescence intensity. The DRAQ-5 data show that the

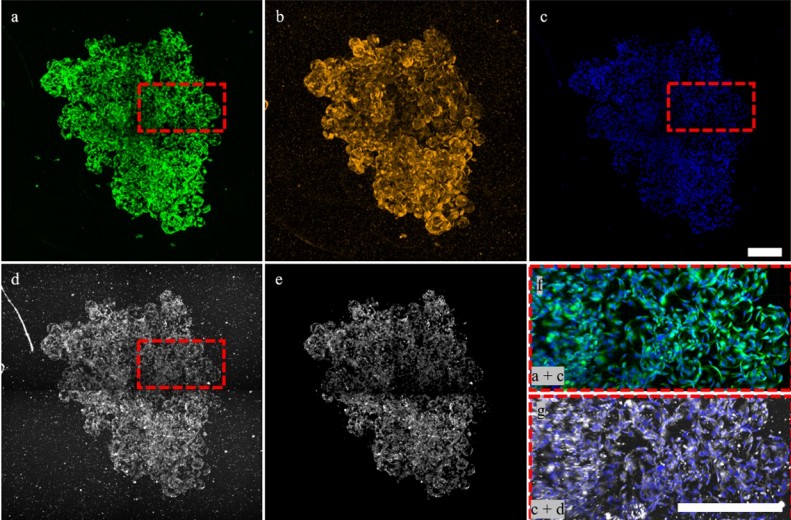

**Fig 3. Large aggregates can be imaged and analyzed *in toto* using fluorescence and label-free LSM.** 2D maximum fluorescence intensity Z-projections of ih-MSCs labeled with **a)** CTG, **b)** DiI, and **c)** DRAQ-5. **(d)** Elastic scattering also allows for visualization of cells within large aggregates. **e)** Small scatterers in agarose and cell debris can be segmented out based on intensity and/or size using Imaris. **f)** CTG + DRAQ-5 merge and **g)** elastic scattering + DRAQ-5 merge at higher resolution. Scale bar = 400 μm.

average cells/microcarrier increased from day 3 to day 7 during both passages, but at a greater rate during passage 4 than passage 7 (Fig 4a). However, there was a lower average cells/microcarrier for both timepoints in passage 7 compared to passage 4. Additionally, larger aggregates were seen in passage 4 than passage 7 at day 7, and the average cells/microcarrier of aggregates > 50 microcarriers was 5.60 and 4.20 for passage 4 day 7 and passage 7 day 7, respectively. The average single cell volumes quantified by CellTracker Green fluorescence and elastic scattering showed similar overall trends (Fig 4b); there was little change in the average cell volume throughout passage 4, but passage 7 cells were larger in volume overall and actually decreased in volume from day 3 to day 7. This study of microcarrier cell growth shows that CellTracker Green and elastic scattering data allow quantification of cell volume. Both fluorescence and elastic scattering modalities showed a linear correlation between total cell volume and nuclear fluorescence-validated cell number at both passages throughout both timepoints (Fig 4c–4f). The modalities showed almost equivalent goodness-of-fit values; 0.98 at passage 4 and 0.93 at passage 7.

## Discussion

Visualization-based monitoring of cytotherapeutic cells during expansion has provided an evidence-based, cost-effective, and minimally-invasive means to assess culture health in real time; however, standard widefield microscopy methods used to evaluate monolayer cultures do not readily translate to 3D microcarrier-based cultures. Our work here is aimed at addressing this need for high-throughput evaluation of cells grown on spherical microcarriers by using fast and photo-gentle lightsheet microscopy combined with image analysis for robust and (semi-) automated analysis. There are two key innovations of this work. First, is the development of an off-line, volumetric, fluorescence-based assay using LSFM and image analysis for direct cell enumeration and cell volume quantification of ihMSC-microcarrier samples that range over an order of magnitude in size: from 100 μm early in culture to > 1 mm later in culture. Second,

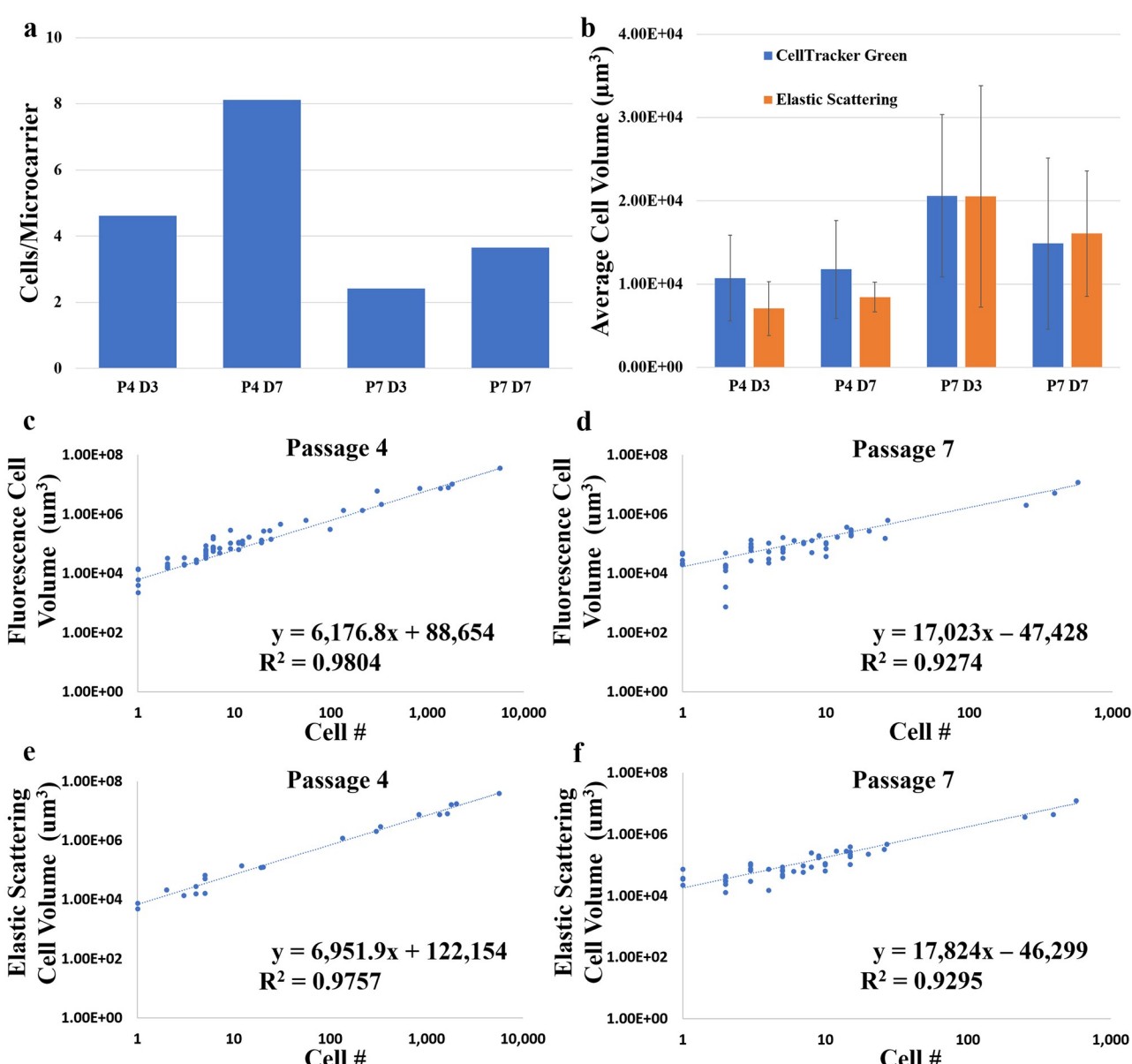

**Fig 4. Optical imaging and semi-automated image analysis enable rapid characterization of microcarrier-expanded cell growth *in toto*. a)** The overall average CPM at each sampled timepoint for all microcarriers sampled including aggregates from the DRAQ-5 data. **b)** The average single cell volume quantified by the fluorescence-based and the elastic scattering-based methods. Data are presented as mean with standard deviation (error bars). The DRAQ-5 labeled cell number versus the total cell volume quantified from the CellTracker Green fluorescence for all **c)** passage 4 and **d)** passage 7 samples. The DRAQ-5 labeled cell number versus the total cell volume quantified by the ELIAS method for all **e)** passage 4 and **f)** passage 7 samples. The linear trends and $R^2$ values are shown.

is the development and demonstration of a non-destructive assay for monitoring of cellular growth via the utilization of esLSM and image analysis that could be further developed into an on-line PAT using microfluidics connected to the bioreactor culture for sampling. In this work, the fluorescence-based assay is used as ground truth data to correlate the elastic scattering-based assay data, but as validated, esLSM can be used for label-free monitoring of microcarrier samples independently of fluorescence-based assays.

These experiments demonstrate, for the first time, direct cell enumeration and cell volume quantification of large microcarrier aggregates, and that non-destructive elastic scattering contrast can be used to monitor microcarrier-bioreactor cell culture growth *in toto*, without the need for cellular detachment, membrane lysing, or exogenous labeling (S6 Fig). The minimal refractive index mismatch between the hydrogel microcarriers and surrounding agarose medium permits high-resolution visualization of cell morphology even throughout large cell-microcarrier aggregates. Similarly, the refractive index differences between the gelMA microcarrier, surrounding agarose, and cell matter cause all 3 classes of objects to scatter at varying intensities and therefore, they can be segmented from each other.

Only a handful of studies have focused on the development of optical imaging systems and methods for studying microcarrier-based cell cultures. Jakob et al. used confocal microscopy and LSFM to image MDCK-II cells on Cytodex 3 microcarriers, but only acquired half microcarrier stacks and required sample rotation to visualize cells along the entire microcarrier surface [63]. The optical projection tomography methodology used in Jakob et al. increases the acquisition time and amount of data needed to accurately reconstruct the 3D cell-microcarrier sample compared to more conventional z-stacks for 3D data reconstruction. Duchi et al. demonstrated that optical sectioning via LSFM enables imaging of small cell-microcarrier clumps, but did not image or analyze clumps or aggregates of more than 5 microcarriers as their study focused on single cell motility and distribution on individual microcarriers [74]. *In situ* microscopy and micro-flow imaging enable direct enumeration of cells attached to microcarriers, but both methods are widefield techniques limited to visualizing the proximal half of the microcarrier and neither has the spatial resolution nor contrast for single cell visualization [43], [48]. Odeleye et al., using a custom *in situ* microscope, was only able to image and analyze the proximal microcarrier surface and struggled to enumerate cells accurately once aggregation began [75]. Microflow imaging was used to broadly characterize microcarrier confluency of single microcarriers and small clumps, but the authors did not investigate the ability to analyze large aggregates [76]. Similarly, the map projection analysis method used by Baradez and Marshall to characterize cell growth on individual microcarriers from confocal microscopy data is likely unfeasible for large cell-microcarrier aggregates with true 3D structure [77]. Imaging cytometry presents an attractive solution for on-line visualization and quantification of cells attached to microcarriers, but few studies have used imaging cytometry characterized cells attached to spherical microcarriers or large cell aggregates [46, 47, 78].

The linear relationship between cell number and cell volume quantified by non-destructive *in toto* ELIAS provides confirmation that this method could be used to image and characterize samples in an aqueous environment. An ELIAS PAT for on-line monitoring of microcarrier cell cultures would further incorporate microfluidic chips and hydraulic flow to sample microcarriers from the bioreactor to the lightsheet for rapid analysis and back to the bioreactor culture [79, 80]. Furthermore, single objective lightsheet systems or oblique plane microscopy could better enable on-line imaging of microfluidic samples as there is only a single sample-facing objective and more space for sample mounting and translation below or above the objective [81, 82]. Additional motivation is presented with development of microfluidic chips composed of a polymer with a refractive-index matched to water which is compatible with the presented ELIAS method [83]. These systems and methods are more complex than traditional sampling and imaging methods, but would enable non-destructive, robust, and automated analysis of microcarrier-based and other suspension cultures. Once a cell culture process for cytotherapy manufacturing is standardized and the trend line between off-line fluorescence-validated cell enumeration, on-line ELIAS-quantified, and cytotherapy product quality is confirmed, deviations from the trend line could indicate issues with the culture health and quality.

As few as 10–20 populated microcarriers may be needed to quantify the average cells per microcarrier throughout a culture [43].

The *in situ* and *in toto* study of large cell aggregates has, up to this point, been minimal due to the limited imaging depth of conventional microscopes and the opacity of large cell-micro-carrier aggregates [75, 77, 84, 85]. Additionally, large aggregates are notoriously difficult to manipulate for study and can cause sampling errors [43, 86]. *In toto* or *in situ* volumetric imaging of these structures would enable analysis of cell morphology, density, and spatial distribution of cell viability throughout an aggregate [30]. Fortunately, the optical sectioning and the decoupled illumination and detection arms of LSM enable imaging deep ($\gg$1 mm) into aggregates. Moreover, the combination of water-dipping objective lens with a long working distance (2 mm), objective lens axial correction collar to fine tune the refractive index mismatch, and hydrogel microcarriers allows for sub-micron resolution volumetric optical imaging of cells attached to spherical microcarriers *in toto*. Because all the pixels in a frame are acquired in parallel, acquisition time can still be relatively short (~1–2 minutes/4D dataset) even for large aggregates >1 mm in depth. The stripe artifacts, which arise from scattering and/or absorption of the illumination beam by small objects such as air bubbles, debris, or un-melted agarose particles, in LSM images can be removed via a number of hardware and image-processing methods, such as digital scanned lightsheet microscopy (DSLM) with pivot scanning or median digital filtering [87]. Future image analysis pipelines could include corrections for uneven illumination, striping artifacts, or photobleaching [88]. Photobleaching, a chemical alteration of fluorophores that render them unable to fluoresce, is not a concern in esLSM as it does not involve an energy transition; however, extra care has to be taken in esLSM to utilize the full dynamic range of the detector without saturation, even with very low incident laser power.

Standard widefield microscopes, routinely used to visualize and qualitatively evaluate cell culture health, integrate photons from in- and out-of-focus planes, making them suitable for thin samples. Dimensionality reduction, by using a single 2D image for volumetric readouts of 3D cell cultures, leads to cell enumeration and segmentation errors that increase with cell growth and cell-microcarrier aggregates [58, 60, 86]. Cell volume could not be quantified from a single 2D projection of the microcarrier. Volumetric microscopy does enable single cell morphological measurements, such as sphericity or nuclear-to-cytoplasmic volume ratio, for low confluency microcarriers where cells do not overlap (S8 Video). More accurate 3D segmentation methods are needed for single cell segmentation and characterization at moderate or high confluency levels where cells overlap. High-throughput LSM enabling single cell morphological monitoring and profiling could be performed with a more robust cell segmentation method for 3D microcarrier cell cultures [46, 89]. In this study, there was a decrease in cell proliferation and an increase in cell size in passage 7 compared to passage 4, which may be due to replicative senescence [90, 91]. Additional image analysis and more frequent culture sampling could be undertaken to see if time-dependent senescence can be visualized and characterized for microcarrier cultures, as has been investigated for monolayer cultures [32, 33, 92]. Although high-throughput single cell morphological cytometry was not performed here, the presented fluorescence- and elastic-scattering cell growth characterization methods preserve all the spatial (3D cell morphology, distribution) information that is lost to cell lysing and detachment (S9 Video). The ability to characterize single cell morphology will better facilitate real-time decision making regarding the health, quality, or futility of a cell culture process.

## Conclusion

We present optical imaging and image analysis methods for direct cell enumeration and cell volume quantification for microcarrier-expanded cells using *in toto* LSFM and esLSM. Both

academic researchers and industry cytotherapy manufacturers would greatly benefit from the ability to monitor and quantify cell culture growth using a non-destructive imaging-based PAT. To the best of our knowledge, this is the first time direct cell enumeration of microcarrier culture aggregates has been reported in the literature.

We used LSM to characterize the entire microcarrier surface, whereas other imaging-based microcarrier growth monitoring methods require either cell membrane lysing or cell-microcarrier detachment, or can only study the proximal half of the microcarrier. Also, we illustrate that our gelMA microcarriers have superior optical imaging capabilities that allow for reliable cell culture monitoring. By incorporating esLSM and hydrogel microcarriers, the ELIAS method presents a strong proof of concept for a non-destructive PAT for monitoring of cytotherapy manufacturing critical parameters. The addition of refractive-index matched microfluidic chips and hydraulic flow would enable on-line ELIAS monitoring of microcarrier-based cell cultures.

## Supporting information

**S1 Fig. Illustration of custom 3D-printed sample chamber for imaging on the Zeiss Lightsheet microscope.** a) The chamber allows for dual-sided lightsheet and trans-illumination. b) The chamber can be scanned in all 3 dimensions and rotated for stitching and precise sample positioning for 3D optical imaging. Not to scale.
(TIF)

**S2 Fig. Image processing workflow for direct cell enumeration and segmentation of cell nuclei using DRAQ-5 fluorescence.** The segmented cell nuclei can be analyzed and classified by, for example, average distance to 3 nearest neighbors.
(TIF)

**S3 Fig. Image processing workflow for cell segmentation and quantification using Cell-Tracker Green data.** Single cells can be classified by volume using Imaris' morphological-based segmentation.
(TIF)

**S4 Fig. Illustration of image processing workflow for label-free cell segmentation and quantification using esLSM data.** Cells scatter at higher intensity values than the microcarrier and surrounding agarose. After the high-pass intensity filter, the voxel filter removes small scatterers or debris in the agarose.
(TIF)

**S5 Fig. Illustration of image processing workflow for microcarrier segmentation and enumeration using esLSM data.** The hydrogel microcarriers scatter less than the cells and surrounding agarose. A low-pass intensity and 90 μm size filter identify individual spherical microcarriers. The *en face* and orthogonal cross-sections permit visualization of the gelMA microcarriers using elastic scattering contrast.
(TIF)

**S6 Fig. Resulting multi-modal microscopy data.** Max intensity projections of a passage 7 day 7 microcarrier using the raw **a)** DRAQ-5, **b)** CellTracker Green, and **c)** elastic scattering data. **d)** Merge of CTG + DRAQ-5 fluorescence projections. **e)** Merge of the segmented microcarrier, CTG (shaded green), and DRAQ-5 (shaded blue) cell regions. **f)** Merge of the segmented microcarrier, elastic scattering (shaded white), and DRAQ-5 cell volumes. Scale bar = 25 μm.
(TIF)

**S1 Video. 3D LSFM rendering of a passage 7 day 7 microcarrier with several cells.** No pre-processing was performed on these data. A representative volume to illustrate the data quality of lightsheet microscopy and hydrogel microcarriers (CellTracker Green (green) and DRAQ-5 (blue)). Gamma correction for visualization of the microcarrier only.
(MP4)

**S2 Video. Z-stack scan of passage 7 day 7 microcarrier using LSFM.**
(MP4)

**S3 Video. 3D esLSM rendering of segmented cells attached to the passage 7 day 7 microcarrier.** Elastic scattering contrast permits visualization of cells attached to spherical hydrogel microcarriers similar to that of fluorescence-based contrast.
(MP4)

**S4 Video. esLSM z-stack scan illustrating the varying scattering intensities of the cells, microcarriers, and surrounding agarose.** The gelMA microcarrier appear as a solid, non-scattering sphere in the esLSM modality. This allows for direct enumeration of microcarriers.
(MP4)

**S5 Video. 3D fluorescence LSM rendering showing a single cell invading the interior of the hydrogel microcarrier.** CellTracker Green (green) and DiI (gold) show the cell process extending into the center of the microcarrier. DRAQ-5 (blue) apparently resolves two elliptical nuclei within the same nuclear envelope.
(MP4)

**S6 Video. 3D esLSM rendering of the same cell invading the core of a microcarrier.** Elastic scattering contrast is able to visualize higher-resolution interesting biological phenomena.
(MP4)

**S7 Video. CellTracker Green and elastic scattering both enable visualization of complex cell networks.** DRAQ-5-labeled nuclei can be used to semi-automatically large aggregates.
(MP4)

**S8 Video. 3D rendering of passage 7 day 7 segmented microcarrier, CTG cytoplasm, and DRAQ-5 nuclei.** No pre-processing performed to segment the cell bodies, nuclei, and microcarrier using CTG, DRAQ-5, and elastic scattering contrast, respectively. These segmented regions are used to quantify cell and nuclear volume, as well as direct cell and microcarrier enumeration.
(MP4)

**S9 Video. 3D rendering of passage 7 day 7 segmented microcarrier, scattering cell body, and DRAQ-5 nuclei.** No pre-processing performed to segment the microcarrier, cell bodies, and cell nuclei using elastic scattering and DRAQ-5 contrast. The elastic scattering data enables visualization and detection of microcarriers as well as quantification of cell volume.
(MP4)

## Acknowledgments

We thank the Texas A&M University Microscopy and Imaging Center Core Facility (RRID: SCR_022128) for providing the Zeiss Z1 Lightsheet microscope, Imaris image analysis software, and the technical assistance for their use.

## Author Contributions

**Conceptualization:** Oscar R. Benavides, Holly C. Gibbs, Roland Kaunas, Carl A. Gregory, Kristen C. Maitland.

**Funding acquisition:** Holly C. Gibbs, Roland Kaunas, Carl A. Gregory, Kristen C. Maitland.

**Investigation:** Oscar R. Benavides, Berkley P. White.

**Supervision:** Alex J. Walsh, Kristen C. Maitland.

**Visualization:** Oscar R. Benavides.

**Writing – original draft:** Oscar R. Benavides.

**Writing – review & editing:** Oscar R. Benavides, Holly C. Gibbs, Berkley P. White, Roland Kaunas, Carl A. Gregory, Alex J. Walsh, Kristen C. Maitland.

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
