## [Decision Letter · Decision Letter 0]

28 Nov 2022

PONE-D-22-26158Volumetric imaging of human mesenchymal stem cells (hMSCs) for non-destructive quantification of 3D cell culture growthPLOS ONE

Dear Dr. Benavides,

Thank you for submitting your manuscript to PLOS ONE. After careful consideration, we feel that it has merit but does not fully meet PLOS ONE’s publication criteria as it currently stands. Therefore, we invite you to submit a revised version of the manuscript that addresses the points raised during the review process.

We look forward to receiving your revised manuscript.

Kind regards,

Kun Chen, Ph.D

Academic Editor

PLOS ONE

Journal Requirements:

Reviewers' comments:

Reviewer's Responses to Questions

**Comments to the Author**

1. Is the manuscript technically sound, and do the data support the conclusions?

Reviewer #1: Partly

Reviewer #2: Yes

Reviewer #3: Yes

2. Has the statistical analysis been performed appropriately and rigorously? 

Reviewer #1: No

Reviewer #2: Yes

Reviewer #3: N/A

3. Have the authors made all data underlying the findings in their manuscript fully available?

Reviewer #1: Yes

Reviewer #2: Yes

Reviewer #3: Yes

4. Is the manuscript presented in an intelligible fashion and written in standard English?

Reviewer #1: No

Reviewer #2: Yes

Reviewer #3: Yes

5. Review Comments to the Author

Reviewer #1: The manuscript “Volumetric imaging of human mesenchymal stem cells (hMSCs) for non-destructive quantification of 3D cell culture growth” by Benavides et al. achieves 3D imaging of human MSCs in gelMA microcarriers. The authors (1) adapted esLSM (elastic scattering light sheet microscopy) imaging and proved that this non-destructive imaging method can quantitatively characterize the hMSCs volume, as they showed that the esLSM readout is comparable to the LSFM reads using fluorescent labeling; (2) the LSFM can achieve large aggregates in toto imaging with single cell resolution.

I recommend publication in PLOS ONE with minor revision, after the authors have addressed the following comments and questions.

Major:

1. The authors probably could better explicate the key innovation of the work. The esLSM is a well-established technique; The presented cell numeration method is mainly based on the commercially available Imaris software. So, is the key point here to apply those existing tools on microcarrier? Also, as [72] has shown that esLSM can achieve (seems to me) qualitatively-comparable-imaging in tumor spheroid, then what makes those gelMA microcarriers (with superior optical capabilities) more challenging? I think it would help a lot if the authors can clarify those points in the Discussion, or somewhere else in the text.

2. The authors probably could better check their data consistency. For instance, the CPM (cell numbers per microcarriers) in Figure 1m and Figure 4a are not the same. Maybe one used Poisson fitting, while the other used a different method?

3. The authors probably could better organize their text. For instance, on Page 10, the authors stated “The elastic scattering signal seems to originate from the cytoplasm, and the nucleus tends to appear as a cavity that exhibits limited signal (Figure 1h).”, then switched to a lot of details in data-analysis topics, then suddenly switched back again by saying “This cytoplasm dominant elastic scattering phenomena has been previously reported [72], and here we similarly show that nuclear-bound fluorescent markers and elastic scattering microscopy provide complimentary information on different cell regions.” Similar problems also occur several times in the Introduction, Discussion, as well as many other places along the text, making the paper a bit hard to read/follow.

4. Similarly, if the key point of using all the fluorescent labeling (except the large aggregates study) is to show the linearity between CellTracker Green reads and esLSM reads (mostly Figure 4), then the authors should make it clear at the very beginning. It is really confusing to have all those off-line imaging while keeping emphasize “non-destructive” on-line method. More generally, the authors could better state or separate the LSFM parts along the text, as it is very hard to find the real focus when jumping between off-line LSFM and on-line elastic scattering.

Minor:

5. Figure 3f, too low resolution.

6. Figures 4d and 4f, hard to see the details in clusters of data points. Probably re-scale (or put a gap in) the axis?

7. Figures 4c-4f, are they D3 or D7?

Reviewer #2: Benavides et al report an optical imaging method could non-destructively visualize the cell morphology and number inside the spherical microcarrier. They also image and enumerate the large aggregates. Light sheet fluorescence microscopy and light sheet tomography are used to image 3D cell morphology.

Cell number is an important parameter in cell culture process and there is a need for quantification of cell number in biopharmaceutical manufacturing. The images of human mesenchymal stem cells inside the spherical microcarrier are showed in the paper. Meanwhile, cell number could be easily counted through analyzing the images by image analysis software.

Reviewer #3: This manuscript is about employing elastic scattering and fluorescence light sheet microscopy to quantify cell volume and cell number. It appears that the authors are demonstrating two techniques in this manuscript –

a) off-line fluorescence-based direct cell enumeration and cell volume quantification method

b) ELIAS method non-destructively monitors microcarrier-bioreactor cell culture growth without the need for cellular detachment,

While these two techniques complement each other, for clarity, the authors should consider separating into two standalone articles. Fluorescence-based imaging could be supporting data for the ELIAS. Thoughtful experiments are validated with sufficient results. This work can be published in PLOS One Journal after a minor revision.

• Figure 1 shows the fluorescence and elastic scattering images. The authors mention that the elastic scattering from cytoplasm is already reported, it is motioned that the new demonstration is elastic scattering from the nucleus.

• Vertical and horizontal scales of 1(m) are hard to read and hence conclude anything.

• Is there a limit to the cell volume size that this detection method can be applied?

• It is mentioned that method is semi-automated – but it is not clear the “automated” part.

• What is the imaging depth of this study? What is the dynamic range - the minimum and maximum cells/microcarrier – that the authors observed in this study. These two points were addressed in the manuscript, but the numbers are for ELIAS and not for this study. Also, instead of spelling out each and every advantage of ELIAS, perhaps authors should consider citing the advantages of the techniques in one or two sentences.

6. PLOS authors have the option to publish the peer review history of their article (what does this mean?). If published, this will include your full peer review and any attached files.

Reviewer #1: No

Reviewer #2: No

Reviewer #3: No

---

## [Author Response · Author response to Decision Letter 0]

4 Jan 2023

Response to Reviewer Comments:

Reviewer #1: 

The manuscript “Volumetric imaging of human mesenchymal stem cells (hMSCs) for non-destructive quantification of 3D cell culture growth” by Benavides et al. achieves 3D imaging of human MSCs in gelMA microcarriers. The authors (1) adapted esLSM (elastic scattering light sheet microscopy) imaging and proved that this non-destructive imaging method can quantitatively characterize the hMSCs volume, as they showed that the esLSM readout is comparable to the LSFM reads using fluorescent labeling; (2) the LSFM can achieve large aggregates in toto imaging with single cell resolution.

We thank the editor and reviewers for their careful consideration of our manuscript and the opportunity to address the concerns.

I recommend publication in PLOS ONE with minor revision, after the authors have addressed the following comments and questions.

Major:

1. The authors probably could better explicate the key innovation of the work. The esLSM is a well-established technique; The presented cell numeration method is mainly based on the commercially available Imaris software. So, is the key point here to apply those existing tools on microcarrier? Also, as [72] has shown that esLSM can achieve (seems to me) qualitatively-comparable-imaging in tumor spheroid, then what makes those gelMA microcarriers (with superior optical capabilities) more challenging? I think it would help a lot if the authors can clarify those points in the Discussion, or somewhere else in the text.

Thank you for the valuable suggestion to better highlight the key innovations of this work. 

Revised Discussion paragraphs: 

“Visualization-based monitoring of cytotherapeutic cells during expansion has provided an evidence-based, cost-effective, and minimally-invasive means to assess culture health in real time; however, standard widefield microscopy methods (brightfield, phase-contrast, and/or fluorescence) used to evaluate monolayer cultures do not readily translate to 3D microcarrier-based cultures. Our work here is aimed at addressing this need for high-throughput evaluation of cells grown on spherical microcarriers by using fast and photo-gentle lightsheet microscopy combined with image analysis for robust and (semi-) automated analysis. There are two key innovations of this work. First, is the development of an off-line, volumetric, fluorescence-based assay using LSFM and image analysis for direct cell enumeration and cell volume quantification of ihMSC-microcarrier samples that range over an order of magnitude in size (100 µm early in culture to over > 1 mm later in culture). Second, is the development and demonstration of a non-destructive assay for monitoring of cellular growth via the utilization of esLSM and image analysis that could be further developed into an on-line PAT using microfluidics connected to the bioreactor culture for sampling. The fluorescence-based assay is used as ground truth data to correlate the elastic scattering-based assay data. 

These experiments demonstrate, for the first time, direct cell enumeration and cell volume quantification of large microcarrier aggregates, and that non-destructive elastic scattering contrast can be used to monitor microcarrier-bioreactor cell culture growth in toto, without the need for cellular detachment, membrane lysing, or exogenous labeling (S6 Fig). The minimal refractive index mismatch between the hydrogel microcarriers and surrounding agarose medium permits high-resolution visualization of cell morphology even throughout large cell-microcarrier aggregates. Similarly, the refractive index differences between the gelMA microcarrier, surrounding agarose, and cell matter causes all 3 classes of objects to scatter at varying intensities and therefore, they can be segmented from each other.”

We disagree with the reviewer that the tumor spheroid images shown in [72] are of the same quality as our images presented in this work. The density of the tumor spheroid samples precludes single cell delineation, and as the authors of [72] point out, only the quadrant (~100 um depth) of the sample nearest the illumination and detection objectives provides “clear imaging.” Our imaging method is application-specific for anchorage-dependent cell-microcarrier samples, and works over a large range of microcarrier sample sizes: we’re able to image single microcarriers (~100 um diameter) such as those shown in Figure 1, or large aggregate structures nearing 2 mm in either XYZ dimension as in Figure 4. Our imaging method is able to accomplish this due to 1) precise refractive index matching of the detection lens correction collar to the gelMA microcarrier refractive index (which we empirically measured and reported in [24]) and 2) dual-sided illumination for more uniform illumination of large aggregate samples. Additionally, our work includes quantitative analysis of esLSM images whereas the authors of [72] only reported the imaging system. The inclusion of [72] was to illustrate 

2. The authors probably could better check their data consistency. For instance, the CPM (cell numbers per microcarriers) in Figure 1m and Figure 4a are not the same. Maybe one used Poisson fitting, while the other used a different method?

Thank you for noting this difference. The CPM data in Figure 1m is for single microcarriers only, whereas the CPM data in Figure 4a is for all microcarriers imaged including aggregates. The inclusion of CPM data from aggregates would dramatically skew the histogram data if included in Figure 1m so instead the aggregate CPM data is weighted and averaged with single microcarriers for the overall average CPM data shown in Figure 4a. 

“The DRAQ-5 volumes were used to monitor the distribution of cells/microcarrier on single microcarriers over time (Figure 1m). Aggregates were excluded from the histograms to avoid weighting against single microcarriers.”

3. The authors probably could better organize their text. For instance, on Page 10, the authors stated “The elastic scattering signal seems to originate from the cytoplasm, and the nucleus tends to appear as a cavity that exhibits limited signal (Figure 1h).”, then switched to a lot of details in data-analysis topics, then suddenly switched back again by saying “This cytoplasm dominant elastic scattering phenomena has been previously reported [72], and here we similarly show that nuclear-bound fluorescent markers and elastic scattering microscopy provide complimentary information on different cell regions.” Similar problems also occur several times in the Introduction, Discussion, as well as many other places along the text, making the paper a bit hard to read/follow.

We appreciate the reviewer pointing this out, and have edited the manuscript to improve readability. 

Revised Results paragraph: The CTG volumes reveal the individual cell morphologies, and there appeared to be an increase in total cell volume from day 3 to day 7 in both passages (Figure 1a-d). The label-free elastic scattering shows similar cell morphology and cell growth over time as the fluorescence data (Figure 1e-lh). This suggests both contrast methods can be used to view cells and quantify cell volume. The DRAQ-5-labeled nuclei data show an increase in cell density from day 3 to day 7 within each passage (Figure 1i-l). A max. of 19 cells per microcarrier was observed at Passage 4 Day 7 using the DRAQ-5 data (Figure 1j). The DRAQ-5 volumes were used to monitor the distribution of cells/microcarrier on single microcarriers over time (Figure 1m). Aggregates were excluded from the histograms to avoid weighting against single microcarriers. Similarly, non-populated microcarriers were not studied. A zero-truncated Poisson distribution was fit to the histograms to account for excluding empty microcarriers from acquisition and analysis [71]. These data suggest a decrease in cells/microcarrier or cell density at passage 7 compared to passage 4 overall. The elastic scattering signal seems to originate from the cytoplasm, and the nucleus tends to appear as a cavity that exhibits little to no elastic scattering signal (Figure 1h). This cytoplasm-dominant elastic scattering phenomena has been previously reported [72], and here we similarly show that nuclear-bound fluorescent markers and elastic scattering microscopy provide complimentary information on different cell regions.

4. Similarly, if the key point of using all the fluorescent labeling (except the large aggregates study) is to show the linearity between CellTracker Green reads and esLSM reads (mostly Figure 4), then the authors should make it clear at the very beginning. It is really confusing to have all those off-line imaging while keeping emphasize “non-destructive” on-line method. More generally, the authors could better state or separate the LSFM parts along the text, as it is very hard to find the real focus when jumping between off-line LSFM and on-line elastic scattering.

We appreciate the reviewer pointing this out, and have edited the manuscript to improve readability. 

Revised Introduction paragraph: Here, we report a proof-of-concept study on volumetric optical imaging and semi-automated image analysis for off-line fluorescence and on-line elastic scattering quantification of cell number and volume of hMSCs cultured on spherical hydrogel microcarriers in toto, without the need for cellular detachment. The off-line fluorescence assay utilizes LSFM and CellTracker Green cytoplasmic and DRAQ-5 nuclear labeling for cell volume quantitation and direct cell enumeration of single microcarriers and large aggregates. This fluorescence assay is the first imaging-based assay to use volumetric data to more accurately characterize the 3D microcarrier cell culture and the first to directly enumerate cells within large aggregates in toto. The on-line esLSM and image analysis (ELIAS) assay is used to quantify cell volume of single microcarriers and aggregates non-destructively. Cell number from the fluorescence assay was correlated to cell volume from the ELIAS assay. The ELIAS assay has the capability to be adopted as an on-line PAT for robust non-destructive monitoring of bioreactor-microcarrier cell culture growth which would improve process and quality control for cytotherapy manufacturing.

Minor:

5. Figure 3f, too low resolution.

The PLOS One editorial portal produced a down-sampled/low-resolution scan of the submitted figure for online publication. A high-resolution figure has been submitted to PLOS One with the original manuscript submission. 

6. Figures 4d and 4f, hard to see the details in clusters of data points. Probably re-scale (or put a gap in) the axis?

We appreciate the reviewer pointing this out, and have plotted the data on the log scale.

7. Figures 4c-4f, are they D3 or D7?

Each figure is from an entire passage, including both D3 and D7 timepoints. Figures 4c and 4e are Passage 4 and Figure 4d and 4f are Passage 7 data points. 

Revised Figure 4 caption: 

a) The overall average CPM at each sampled timepoint for all microcarriers sampled including aggregates from the DRAQ-5 data. b) The average single cell volume quantified by the fluorescence-based and the elastic scattering-based methods. Data are presented as mean with standard deviation (error bars). The DRAQ-5 labeled cell number versus the total cell volume quantified from the CellTracker Green fluorescence for all c) passage 4 and d) passage 7 samples. The DRAQ-5 labeled cell number versus the total cell volume quantified by the ELIAS method for all e) passage 4 and f) passage 7 samples. The linear trends and R2 values are shown.

Revised Results paragraph section: This study of microcarrier cell growth shows that CellTracker Green and elastic scattering data allow quantification of cell volume. Both fluorescence and elastic scattering modalities showed a linear correlation between total cell volume and nuclear fluorescence-validated cell number at both passages throughout both timepoints (Figure 4c-f).

Reviewer #2: Benavides et al report an optical imaging method could non-destructively visualize the cell morphology and number inside the spherical microcarrier. They also image and enumerate the large aggregates. Light sheet fluorescence microscopy and light sheet tomography are used to image 3D cell morphology.

Cell number is an important parameter in cell culture process and there is a need for quantification of cell number in biopharmaceutical manufacturing. The images of human mesenchymal stem cells inside the spherical microcarrier are showed in the paper. Meanwhile, cell number could be easily counted through analyzing the images by image analysis software.

Reviewer #3: This manuscript is about employing elastic scattering and fluorescence light sheet microscopy to quantify cell volume and cell number. It appears that the authors are demonstrating two techniques in this manuscript –

a) off-line fluorescence-based direct cell enumeration and cell volume quantification method

b) ELIAS method non-destructively monitors microcarrier-bioreactor cell culture growth without the need for cellular detachment,

While these two techniques complement each other, for clarity, the authors should consider separating into two standalone articles. Fluorescence-based imaging could be supporting data for the ELIAS. Thoughtful experiments are validated with sufficient results. This work can be published in PLOS One Journal after a minor revision.

• Figure 1 shows the fluorescence and elastic scattering images. The authors mention that the elastic scattering from cytoplasm is already reported, it is motioned that the new demonstration is elastic scattering from the nucleus.

We appreciate the reviewer pointing this out, and have edited the manuscript to improve readability. The data suggests that the elastic scattering signal is predominately from the cytoplasm, and that there is little to no scattering signal from the nucleus. 

Revised Results paragraph: The elastic scattering signal seems to originate from the cytoplasm, and the nucleus tends to appear as a cavity that exhibits little to no elastic scattering signal (Figure 1h). This cytoplasm-dominant elastic scattering phenomena has been previously reported [72], and here we similarly show that nuclear-bound fluorescent markers and elastic scattering microscopy provide complimentary information on different cell regions.

• Vertical and horizontal scales of 1(m) are hard to read and hence conclude anything.

The PLOS One editorial portal produced a down-sampled/low-resolution scan of the submitted figure for online publication. A high-resolution figure has been submitted with increased font size for Figure 1(m). 

• Is there a limit to the cell volume size that this detection method can be applied?

In principle, the spatial resolution of the imaging method determines the smallest cell size that can be resolved and therefore detected. With a 20X 1.0 NA detection objective lens, 1.6X variable zoom, and 6.5 µm x 6.5 µm actual camera pixel size, the lateral resolution is determined by the effective (6.5 µm / (20*1.6)) pixel size which is 0.20 µm. With 10X 0.2 NA illumination objective lenses, the axial resolutions, estimated by the lightsheet thickness, for 488 nm and 638 nm illumination are ~2.07 µm and ~2.70 µm, respectively. Following Nyquist sampling principles, the smallest resolvable cell would be => 0.40 µm in a lateral dimension. 

In theory, there is no upper limit to the microcarrier aggregate size that can be imaged using the lowest magnification on the lightsheet and tiling or stitching of multiple field of views together. As is evident in Figure 3, however, there is a loss of both fluorescence and elastic scattering signal intensity towards the center of aggregates due to the increased illumination and imaging depth required for light penetration, although this could be overcome with post-acquisition image processing methods such as shade corrections. The detection lens has a 2 mm working distance, so the max. sample thickness would be 4 mm. 

• It is mentioned that method is semi-automated – but it is not clear the “automated” part.

As opposed to traditional cell counting where a user manually searches the imaged volume for cell nuclei, a posteriori knowledge is used as we expect the cell nuclei to be at least 10 µm across in the lateral dimensions, and use this as a minimum spot size in the Imaris ‘Spot’ algorithm to automatically detect and segment any objects larger than 10 µm. This excludes most cell debris on the microcarrier surface. Then, an intensity threshold is manually set to further exclude non-nuclei objects as cell nuclei fluoresce at higher intensities than cell debris (S2 Figure). Similarly, for microcarrier counting, a 90 µm size filter is input in the Imaris ‘Spot’ algorithm to automatically detect and enumerate microcarriers within an aggregate (S5 Figure).

Revised Methods paragraph:

Imaris image analysis software was used to view and analyze the 3D volumes. For direct cell enumeration, the ‘Spot’ function was used on the DRAQ-5 volumes with an object size filter of 10 µm for automatic detection of cell nuclei, followed by a manual high-pass intensity threshold to further segment out cell debris and ultimately enumerate only cell nuclei (S2 Fig). For quantification of cell volume, the ‘Surface’ function was used on the CTG volumes with a manual high-pass intensity threshold and size filter to remove cell debris from quantification (S3 Fig.). No preprocessing of the data was required for cell segmentation as the gelMA microcarriers produce little background signal [24]. For enumerating microcarriers, the ‘Surface’ function was used on the CTG volume with a low-pass intensity-based threshold to create a solid object. Then, the ‘Spot’ function with a 90 µm size filter was used to automatically enumerate individual spherical microcarriers.

• What is the imaging depth of this study? What is the dynamic range - the minimum and maximum cells/microcarrier – that the authors observed in this study. These two points were addressed in the manuscript, but the numbers are for ELIAS and not for this study. Also, instead of spelling out each and every advantage of ELIAS, perhaps authors should consider citing the advantages of the techniques in one or two sentences.

We appreciate the reviewer asking for additional study details. The maximum imaging depth achievable for the given optics in the Zeiss Z1 lightsheet is 2 mm given the 2 mm working distance of the detection objective lens. With 180-degree sample rotation and post-acquisition stitching, this could be doubled to image an effective depth of 4 mm. 

Empty microcarriers were excluded from the study, although they were seen at all 4 timepoints with varying frequency. The maximum number of cells seen on a single microcarrier was 19 cells at Passage 4 Day 7 (Figure 1j). Alternatively, the largest aggregate seen had 5,673 cells and 1,754 microcarriers within it at Passage 4 Day 7 (Figure 3). 

Results sentence inclusion: A max. of 19 cells per microcarrier was observed at Passage 4 Day 7 using the DRAQ-5 data (Figure 1j).

---

## [Decision Letter · Decision Letter 1]

14 Feb 2023

Volumetric imaging of human mesenchymal stem cells (hMSCs) for non-destructive quantification of 3D cell culture growth

PONE-D-22-26158R1

Dear Dr. Benavides,

We’re pleased to inform you that your manuscript has been judged scientifically suitable for publication and will be formally accepted for publication once it meets all outstanding technical requirements.

Kind regards,

Kun Chen, Ph.D

Academic Editor

PLOS ONE

Additional Editor Comments (optional):

Reviewers' comments:

Reviewer's Responses to Questions

**Comments to the Author**

1. If the authors have adequately addressed your comments raised in a previous round of review and you feel that this manuscript is now acceptable for publication, you may indicate that here to bypass the “Comments to the Author” section, enter your conflict of interest statement in the “Confidential to Editor” section, and submit your "Accept" recommendation.

Reviewer #1: All comments have been addressed

Reviewer #3: All comments have been addressed

2. Is the manuscript technically sound, and do the data support the conclusions?

Reviewer #1: Yes

Reviewer #3: (No Response)

3. Has the statistical analysis been performed appropriately and rigorously? 

Reviewer #1: Yes

Reviewer #3: (No Response)

4. Have the authors made all data underlying the findings in their manuscript fully available?

Reviewer #1: Yes

Reviewer #3: (No Response)

5. Is the manuscript presented in an intelligible fashion and written in standard English?

Reviewer #1: Yes

Reviewer #3: (No Response)

6. Review Comments to the Author

Reviewer #1: The authors have adequately addressed all my comments. The paper is ready for PLOS ONE publication.

Reviewer #3: Although authors have addressed all my queries, I suggest them to add few sentences about "limit to the cell volume size that this detection method can be applied" in the manuscript. It will help readers and future researchers.

7. PLOS authors have the option to publish the peer review history of their article (what does this mean?). If published, this will include your full peer review and any attached files.

Reviewer #1: No

Reviewer #3: No

---

## [Editor Report · Acceptance letter]

17 Mar 2023

PONE-D-22-26158R1 

Volumetric imaging of human mesenchymal stem cells (hMSCs) for non-destructive quantification of 3D cell culture growth 

Dear Dr. Benavides:

I'm pleased to inform you that your manuscript has been deemed suitable for publication in PLOS ONE. Congratulations! Your manuscript is now with our production department. 

Kind regards, 

on behalf of

Dr. Kun Chen 

Academic Editor

PLOS ONE